# TrackIME: Enhanced Video Point Tracking via Instance Motion Estimation

**Seong Hyeon Park**[1]     **Huiwon Jang**[1]     **Byungwoo Jeon**[1]     **Sukmin Yun**[2]

**Paul Hongsuck Seo**[3]     **Jinwoo Shin**[1]

[1]KAIST     [2]Hanyang University ERICA     [3]Korea University

{seonghyp, huiwoen0516, imbw2024, jinwoos}@kaist.ac.kr
sukminyun@hanyang.ac.kr     phseo@korea.ac.kr

## Abstract

Tracking points in video frames is essential for understanding video content. However, the task is fundamentally hindered by the computation demands for brute-force correspondence matching across the frames. As the current models down-sample the frame resolutions to mitigate this challenge, they fall short in accurately representing point trajectories due to information truncation. Instead, we address the challenge by pruning the search space for point tracking and let the model process only the important regions of the frames without down-sampling. Our first key idea is to identify the object instance and its trajectory over the frames, then prune the regions of the frame that do not contain the instance. Concretely, to estimate the instance's trajectory, we track a group of points on the instance and aggregate their motion trajectories. Furthermore, to deal with the occlusions in complex scenes, we propose to compensate for the occluded points while tracking. To this end, we introduce a unified framework that jointly performs point tracking and segmentation, providing synergistic effects between the two tasks. For example, the segmentation results enable a tracking model to avoid the occluded points referring to the instance mask, and conversely, the improved tracking results can help to produce more accurate segmentation masks. Our framework can be easily incorporated with various tracking models, and we demonstrate its efficacy for enhanced point tracking throughout extensive experiments. For example, on the recent TAP-Vid benchmark, our framework consistently improves all baselines, *e.g.*, up to 13.5% improvement on the average Jaccard metric. The project url is https://trackime.github.io/.

## 1  Introduction

Obtaining accurate point trajectories over the video frames is crucial for understanding complex dynamics in video data, a necessity for advanced spatial-temporal tasks like action recognition [2], novel-view rendering [3], video frame prediction/interpolation [4], and video depth estimation [5]. Recently, video point tracking task [6, 7, 8, 9, 10] has witnessed rapid progress, which aims to predict the trajectory and visibility[1] of a given query point, proving long-term trajectories robust to partial occlusions of objects in real video scenes.

Despite their success, we find current point tracking models are fundamentally challenged by an excessive computation demand since the task requires brute-force comparisons over every spatial location in every frame in a given video. As a result, to meet the computation constraints, the models

---

[1]The confidence whether the trajectory is visible in each frame; *i.e.*, the point is not out-of-frame and not occluded by different objects.

38th Conference on Neural Information Processing Systems (NeurIPS 2024).

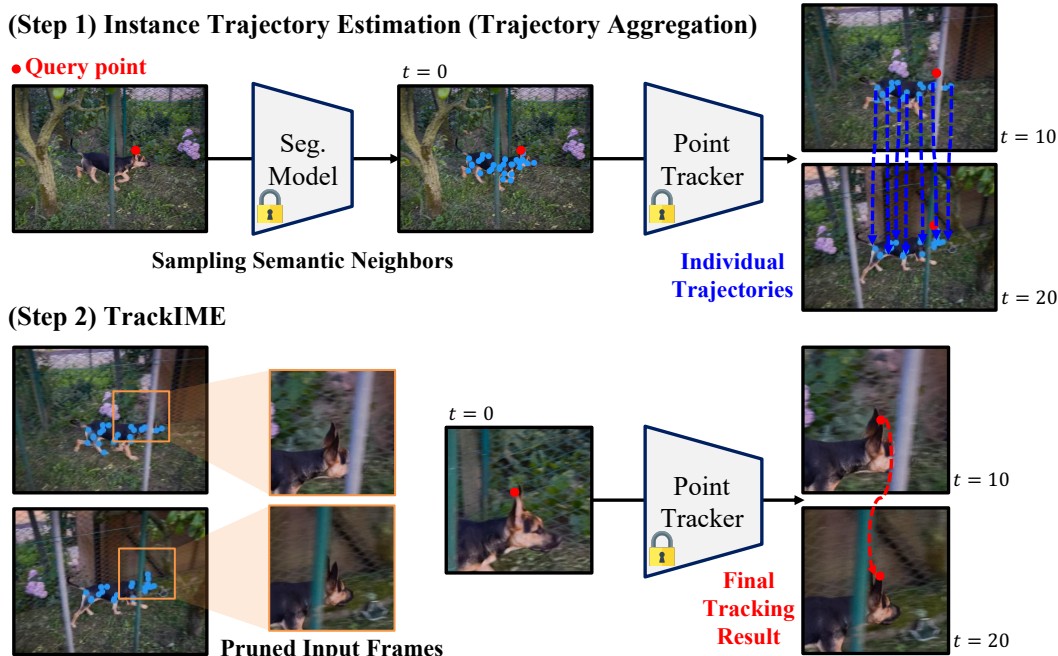

Figure 1: **The workflow of TrackIME.** Our framework enhances point tracking by pruning the search space, along the instance trajectory in video frames. To estimate the instance trajectory, our framework utilizes the point tracking results for a group of points (blue lines) on top of the object instance predicted by segmentation model (*e.g.*, SAM [1]) and aggregate their individual trajectories.

down-sample their tracking resolutions, sacrificing detailed visual features, which eventually leads to sub-optimal tracking accuracy and triggers tracking failures on intricate object parts. In this regard, we pursue the direction of pruning the excessive search space for point tracking, so that models can avoid the down-sampling and focus only on important regions maintaining detailed visual features, *e.g.*, the object instance masks the query point lies in.

In this paper, we introduce TrackIME: Enhanced Video Point Tracking via Instance Motion Estimation that focuses on the region occupied by the object instance that the queried point lies in and guides point tracking models to prune the video frames along the instance's motion trajectory. Here, to obtain the instance trajectory, we first produce the instance mask for a given query point by utilizing the recent segmentation foundation models, *e.g.*, segment anything (SAM) [1], where these foundation models show strong generalization performance to different objects/scenes and we find resulting instance masks in quality are readily available. Then, given the instance mask, we sample a set of points and aggregate their tracking results as the estimate of the instance trajectory.[2]

Furthermore, to deal with the occlusions in complex video scenes, we propose a unified framework that jointly performs the point tracking and video segmentation, where it re-samples the occluded points by referring to the instance mask. We note that our framework provides synergistic effects for both tasks, *i.e.*, the point tracking results assisted by the segmentation can conversely bolster the quality of segmentation. Consequently, although our primary focus is on the advances in point tracking, our method can also demonstrate improved segmentation results than the baselines (see Section 4.2 for details).

Through the experiments on the TAP-Vid point tracking benchmark [11], we demonstrate the effectiveness of TrackIME by incorporating it with different point tracking models such as TAPIR [6]. For example, in the DAVIS scenes [12] evaluating the point tracking for dynamic objects, our method achieved up to 13.5% relative improvement (57.5 → 65.3 with TAPIR) in terms of the average Jaccard (AJ) metric. Moreover, as our framework allows pruning non-instance regions for point tracking

---

[2]Intuitively, the instance as a group of points moves together even if a fine-grained motion of individual points may differ. Hence, we track multiple points on the same instance, and then aggregate their trajectories as the instance motion, which we eventually utilize for pruning video frames.

models, the efficacy of our method stands out even more when evaluated on more harsh standards, *e.g.*, the 1-pixel error threshold, where the conventional metrics allow up to 16-pixel errors when judging the prediction to be correct.

## 2 Method

In this section, we describe the detailed procedure of our TrackIME framework and its application to video point tracking. Specifically, in Section 2.1, we describe the formulation for instance trajectory estimation, which is based on the video point tracking of the query points found by the foundation segmentation model [1].

Next, in Section 2.2, we present the detailed formulation of TrackIME given the instance trajectory, which prunes unimportant regions in the video frame and achieves boosted point tracking performances.

As for the data notations, we denote vectors with $N$ elements as bold letters $\boldsymbol{x} := [\boldsymbol{x}_1; \boldsymbol{x}_2; ...; \boldsymbol{x}_N]$, tensors with $N$ arrays as bold capital letters $\mathsf{X} := [\mathsf{X}_1; \mathsf{X}_2; ...; \mathsf{X}_N]$, where the subscripts represent the indexed scalars or arrays. Otherwise, every non-bold symbol is scalar. We also introduce the superscripts, *e.g.*, $\boldsymbol{x}^{(\mathtt{q})}$, when denoting there is special semantics for a data, such as the query point.

Finally, when making binary classifications based on probability (or normalized confidence) values, we use threshold 0.5; nevertheless, the values are hyperparameters and can be altered in practice.

### 2.1 Instance Trajectory Estimation

In this section, we provide the definition of the instance trajectory and procedures to obtain it, such as sampling a group of points on the instance, trajectory aggregation, and the point re-sampling modules.

**Video point tracking**. Let $\mathsf{I} \in \mathbb{R}^{L \times H \times W \times 3}$ be the tensor of video frames, where $L$ denotes the time duration and $(H \times W)$ denotes the image size, and let $\boldsymbol{p}^{(\mathtt{q})} \in \mathbb{R}^2$ be the spatial coordinates of the query point. Typically, we consider the query in the initial frame hence we do not denote the time index of the query point for clarity. Given the video $\mathsf{I}$ and the query point $\boldsymbol{p}^{(\mathtt{q})}$, we consider a point tracking model $\texttt{Tracker}$ that predicts the query trajectory $\boldsymbol{T}^{(\mathtt{q})} \in \mathbb{R}^{L \times 2}$ and the probability of being visible $\boldsymbol{o}^{(\mathtt{q})} \in (0, 1)^L$ over the entire set of frames,

$$(\boldsymbol{T}^{(\mathtt{q})}, \boldsymbol{o}^{(\mathtt{q})}) := \texttt{Tracker}(\boldsymbol{p}^{(\mathtt{q})}, \mathsf{I}). \tag{1}$$

Here, one might utilize Equation (1) as the simplest representation of the instance motion trajectory. However, modeling the instance motion solely using the query point has critical shortcomings. For example, when the instance is partially occluded by other objects, the trajectory of the query point may no longer exist (see Section 4.1 for the ablation study).

To address this challenge, we propose to sample additional tracking points automatically. Specifically, our idea is to identify the instance mask of the query point so that extra query points can be added from the mask.

**Sampling points on the instance**. Let $\mathsf{M}_0 \in (0, 1)^{H \times W}$ denote the segmentation mask that represents the object instance associated with the query point $\boldsymbol{p}^{(\mathtt{q})}$. Given this mask, we sample a group of points on the instance,

$$\mathcal{N}(\boldsymbol{p}^{(\mathtt{q})}) := \{\boldsymbol{p}^{(n_0)}, \ldots, \boldsymbol{p}^{(n_S)}\}, \tag{2}$$

which we refer to it as the *semantic neighbors* of $\boldsymbol{p}^{(\mathtt{q})}$. We note that $S$ is the number of sampled points, where the query point is also counted as its semantic neighbor, *i.e.*, $\boldsymbol{p}^{(n_0)} \equiv \boldsymbol{p}^{(\mathtt{q})}$.

For each semantic neighbor point, we employ $\texttt{Tracker}$ in Equation (1) to produce its trajectory and visibility, $(\boldsymbol{T}^{(n_i)}, \boldsymbol{o}^{(n_i)}) := \texttt{Tracker}(\boldsymbol{p}^{(n_i)}, \mathsf{I})$,[3] and pass it to the trajectory aggregation module. Since the query point also participate in our tracking procedure, the total effective number of points would be $S + 1$. For example, we choose $S + 1 = 32$ in our main experiments discussed in Section 2.

---

[3]In practice, we batch-process a set of multiple points simultaneously.

**Trajectory aggregation.** We produce an instance motion trajectory by aggregating the tracking results of the semantic neighbors. Specifically, we consider the velocity, $\Delta \boldsymbol{T}_t^{(n_i)} := \boldsymbol{T}_t^{(n_i)} - \boldsymbol{T}_{t-1}^{(n_i)}$, and calculate the weighted average:

$$\Delta \bar{\boldsymbol{T}}_t^{(\mathtt{q})} := \sum_{\left(\boldsymbol{o}_t^{(n_i)} \geq 0.5\right)} \frac{\boldsymbol{o}_t^{(n_i)} \cdot \Delta \boldsymbol{T}_t^{(n_i)}}{\sum_{\left(\boldsymbol{o}_t^{(n_j)} \geq 0.5\right)} \boldsymbol{o}_t^{(n_j)}}. \tag{3}$$

In Equation (3), we note that velocities are aggregated only if the points are classified visible ($\boldsymbol{o}_t^{(n_i)} \geq 0.5$), and the visibility acts as the aggregation weight. Finally, we accumulate the aggregated velocity starting from $\bar{\boldsymbol{T}}_0^{(\mathtt{q})} := \boldsymbol{p}^{(\mathtt{q})}$, to obtain the instance motion trajectory,

$$\bar{\boldsymbol{T}}_t^{(\mathtt{q})} := \bar{\boldsymbol{T}}_{t-1}^{(\mathtt{q})} + \Delta \bar{\boldsymbol{T}}_t^{(\mathtt{q})}. \tag{4}$$

**Instance mask.** In order to identify the instance mask, we employ the recent foundation segmentation model, *e.g.*, Segment Anything Model (SAM) [1], and prompt the model with the query point $\boldsymbol{p}^{(\mathtt{q})}$, to produce the pixel-wise confidence representing the object instance indicated by the query point. We denote this function as Seg,

$$\mathbf{M}_0 := \mathtt{Seg}(\boldsymbol{p}^{(\mathtt{q})}, \mathbf{I}_0) \in (0, 1)^{H \times W}. \tag{5}$$

Given the mask $\mathbf{M}_0$, we employ a weighted sampling for the semantic neighbors. Specifically, we encode the sampling weights with the distance transform (DT) [13, 14] to the mask's region with positive classifications,

$$\mathbf{W}_0 := \mathtt{DT}\left(\mathbf{1}[\mathbf{M}_0 \geq 0.5]\right). \tag{6}$$

In this way, the points near the mask's contour are preferred, which we find efficiently represent the object instance because the contour is approximately linearly proportional to the mask's radius.

**Point re-sampling for robustness to occlusion**. Occlusions are common in real video frames, due to dynamic objects and the camera's motion. In the extreme case, Equation (3) can become degenerate when all semantic neighbors are invisible in future frames $t > 0$. Therefore, maintaining a sufficient number of visible tracking points is crucial, and we tackle this issue by re-sampling occluded points from the instance mask jointly predicted while point tracking.

In a nutshell, whenever we find a certain semantic neighbor point $\boldsymbol{p}^{(n_i)}$ becomes invisible at time $t'$ and does not show up again ($\boldsymbol{o}_t^{(n_i)} < 0.5$ for $t \geq t'$), we query the segmentation model with the tracking results of other semantic neighbors to obtain a new mask to re-sample the occluded point:

$$\mathbf{M}_{t'}^{(n_j)} := \mathtt{Seg}(\boldsymbol{T}_{t'}^{(n_j)}, \mathbf{I}_{t'}) \in (0, 1)^{H \times W}. \tag{7}$$

However, they could also have been affected by occlusions (*e.g.*, when $\boldsymbol{o}_{t'}^{(n_j)}$ is close to the threshold 0.5), or by the severe errors in the trajectory $\boldsymbol{T}_{t'}^{(n_j)}$ due to sub-optimal tracking performance of Equation (1). Hence, predicting segmentation with these points in a naive way can lead to erroneous masks being produced.

To address this problem, our key idea is to aggregate the group of segmentation masks. Specifically, we collect individual masks by Equation (7), then apply a weighted average of the positive classifications,

$$\bar{\mathbf{M}}_{t'} := \sum_{\left(\boldsymbol{o}_{t'}^{(n_i)} \geq 0.5\right)} \frac{\boldsymbol{o}_{t'}^{(n_i)} \cdot \mathbf{1}[\mathbf{M}_{t'}^{(n_i)} > 0.5]}{\sum_{\left(\boldsymbol{o}_{t'}^{(n_j)} \geq 0.5\right)} \boldsymbol{o}_{t'}^{(n_j)}} \in (0, 1)^{H \times W}. \tag{8}$$

We find the mask produced by Equation (8) reflects the confidence of each segmentation mask, as well as the visibility of the associated point, and refer to it as the *mixture of segmentation distributions*, where the value in each index represents the segmentation probability of the queried object.

Based on the constructed mixture of segmentation distributions, we obtain the sampling weight in similar manner to Equation (6) as, $\mathbf{W}_{t'} := \text{DT}(\mathbf{1}[\bar{\mathbf{M}}_{t'} \geq r])$, where the threshold $r \in [0, 1)$ is set much smaller than the standard $0.5$.[4] This is because we should allow the confident partial segmentation distributions, but ignore the unconfident noise segmentation distributions.

Finally, we re-sample the additional points with $\mathbf{W}_{t'}$ as the sampling weight. They replace the occluded points for the instance trajectory estimation in subsequent frames $t > t'$. We execute this procedure during the tracking, which ensures that a sufficient number of visible points participate in Equations (3) and (4). For example, we set it to be the same as the number of initial semantic neighbors $S$.

## 2.2 TrackIME: Enhanced Video Point Tracking via Instance Motion Estimation

In this section, we describe our enhanced point tracking, which prunes the search spaces in frames and produces more accurate tracking results.

**Search space pruning.** Given the instance trajectory in Equation (4), we now aim to utilize it for pruning the search space. Specifically, we prune unimportant non-instance regions, by sampling each frame around the $(H_0 \times W_0)$ regions centered at the aggregated trajectory,

$$\mathbf{I}^{(q)} := \text{Prune}(\mathbf{I}, \bar{\boldsymbol{T}}^{(q)}, H_0, W_0) \in \mathbb{R}^{L \times H_0 \times W_0 \times 3}. \tag{9}$$

We note that the sizes $(H_0 \times W_0)$ are set to be close to the down-sampling resolution considered by a tracking model (*e.g.*, $(256 \times 256)$ for TAPIR [6]) so that the information loss is minimized.

Given the frames with pruned search spaces, we execute `Tracker` again to produce the enhanced tracking outputs. Also, for convenience, we abstract the entire process of the instance trajectory estimation (Section 2.1), the pruning (Equation (9)) and the tracking into a function `TrackerHD`,

$$\begin{aligned} (\boldsymbol{T}^{(\text{HD})}, \boldsymbol{o}^{(\text{HD})}) &:= \text{TrackerHD}(\boldsymbol{p}^{(q)}, \mathbf{I}, H_0, W_0) \\ &:= \text{Tracker}(\boldsymbol{p}^{(q)}, \mathbf{I}^{(q)}). \end{aligned} \tag{10}$$

We note that the feature resolutions inside the tracking model are not modified, therefore the computational complexity does not increase.

**Progressive inference.** To achieve a further boost in the tracking performance, we can additionally use a progressive inference structure. Formally, we consider a collection of $K$ different `TrackerHD` models equipped with different pruning sizes $(H_k, W_k)$:

$$\left[\boldsymbol{T}_1^{(\text{HD})}; ...; \boldsymbol{T}_K^{(\text{HD})}\right] \text{ and } \left[\boldsymbol{o}_1^{(\text{HD})}; ...; \boldsymbol{o}_K^{(\text{HD})}\right], \tag{11}$$

where $\boldsymbol{T}_k^{(\text{HD})} \in \mathbb{R}^{L \times 2}$ and $\boldsymbol{o}_k^{(\text{HD})} \in (0, 1)^L$ denotes the outputs of the $k$-th `TrackerHD` model.

This progressive structure can boost the tracking performance in two ways. The first is utilizing a past $k$-th `TrackerHD` as the tracking model that estimates the instance trajectory for the next $(k + 1)$-th `TrackerHD`. In this way, the pruning is guided by a more accurate trajectory estimate. The second is that these $K$ tracking results can be aggregated to produce the final trajectory. Specifically, we aggregate based on the visibility, in a similar manner to Equations (3) and (8):

$$\boldsymbol{T}^{(\text{Final})} := \sum_{k=1}^{K} \frac{\boldsymbol{o}_k^{(\text{HD})} \odot \boldsymbol{T}_k^{(\text{HD})}}{\sum_{l=1}^{K} \boldsymbol{o}_l^{(\text{HD})}}, \tag{12}$$

where $\odot$ indicates the element-wise product. This aggregation allows processing multiple scales in visual features, which can enhance the generalization performance of vision models [15, 16]. We note that the visibility predictions are averaged over the $K$ predictions.

---

[4]For example, we choose $r = 0.1$ for our main experiments discussed in Section 4.

Table 1: **The evaluation of point tracking performance for dynamic objects.** We benchmark the quality of point tracking in DAVIS [12] videos with the point annotations provided by TAP-Net [11]. We note that TrackIME is incorporated with TAPIR point tracker [6].

| Method | First Query | | | | | Strided Query | | | | |
|---|---|---|---|---|---|---|---|---|---|---|
| | $J_1$ | AJ | $\delta_1^x$ | $\delta_{avg}^x$ | OA | $J_1$ | AJ | $\delta_1^x$ | $\delta_{avg}^x$ | OA |
| TAPNet [11] | 20.7 | 51.6 | 30.1 | 63.8 | 79.8 | 25.3 | 56.5 | 36.3 | 68.2 | 82.6 |
| PIPS2 [9] | 19.6 | 46.6 | 35.8 | 69.4 | 80.3 | 6.9 | 52.8 | 14.2 | 65.8 | 83.5 |
| TAPIR [6] | 23.0 | 57.5 | 34.3 | 70.5 | 84.4 | 28.1 | 62.8 | 41.0 | 75.1 | 87.7 |
| CoTracker [7] | 28.3 | 60.8 | 43.5 | 76.1 | 86.0 | 34.9 | 64.3 | 50.9 | 78.9 | **89.1** |
| OmniMotion [8] | 21.5 | 52.6 | 39.1 | 68.1 | 85.4 | 30.1 | 55.6 | 45.1 | 70.3 | 88.9 |
| **TrackIME** | **35.4** | **65.3** | **48.2** | **78.6** | **86.5** | **41.9** | **69.3** | **55.0** | **81.4** | 89.0 |

# 3 Related Work

**Optical Flow.** Optical flow deals with the dense computation of instantaneous motion patterns between two given video frames. Starting with the pioneering work of applying neural networks for motion estimation [17, 18], the seminal works such as DCFlow [19], PWC-Net [20] and RAFT [21] introduced the concept of dense correspondence matching between pairs of image patches. Despite their success, the optical flow's inherent limitations incapable of modeling trajectories and occlusions triggered the recent progress in the point tracking methods.

**Point Tracking.** In essence, point tracking attempts to find the long-term point correspondences over the entire video frames, and model the occlusions and trajectories. The current models in this domain, such as PIPs [22], TAPNet [11], TAPIR [6], CoTracker [7], and OmniMotion [8] has led rapid progress, with advanced neural architectures [6, 7] or test-time optimizations [8]. However, they are fundamentally hindered by the excessive search space for correspondence matching over the entire frames. Our focus is to address this issue by pruning the search space, where our method can be readily incorporated with these baselines.

**Instance Segmentation.** Recently, the important advancement within image segmentation has been the introduction of segment anything (SAM) [1]. SAM is specifically designed to perform image segmentation by general point prompts and exhibits an impressive capacity for class-agnostic segmentation. Specifically, in the context of point tracking, SAM serves as a valuable resource by generating segmentation masks for the object instance indicated by the query point. We also note the line of zero-shot video segmentation [23, 24, 25, 26, 27, 28, 29, 30]. Specifically, the recent SAM-PT [30] focuses on bolstering video segmentation based on point tracking, which is fundamentally different from our work; our primary goal is obtaining better point tracking, while that for SAM-PT is for better segmentation. Nevertheless, our method provides synergistic effects for both tasks, and even outperforms SAM-PT for segmentation tasks (see Table 4).

# 4 Experiments

In this section, we demonstrate the effectiveness of the proposed TrackIME on point tracking tasks and the downstream video object segmentation.

In Section 4.1, we focus on the point tracking tasks. Specifically, we first experiment the efficacy of the instance motion trajectory estimation and our search space pruning technique for point tracking by measuring the performance in video scenes that capture dynamic objects.

Next, we verify the universality of our method to different point tracking models and find whether it can provide general performance improvements when incorporated into the five recent baselines, *e.g.*, TAPNet [11], PIPS2 [9], CoTracker[7], OmniMotion[8] and TAPIR [6].

In the ablation study, we validate the effect of each component, namely the trajectory aggregation, the search space pruning, and the progressive inference modules described in Section 2.

Table 2: **Universality of TrackIME with different point tracking models.** We incorporate recent point tracking model baselines [6, 7, 8, 9, 11] with our method, and benchmark its performance on DAVIS [12], RGBStacking [33], and Kinetics [34]. †: the underlined results are obtained with subsets of RGBStacking and Kinetics datasets due to a large optimization cost for the OmniMotion [8].

| Method | DAVIS | | RGBStacking | | Kinetics | |
|---|---|---|---|---|---|---|
| | AJ | $\delta^x_{\text{avg}}$ | AJ | $\delta^x_{\text{avg}}$ | AJ | $\delta^x_{\text{avg}}$ |
| TAPNet [11] | 51.6 | 63.8 | 56.5 | 79.0 | 49.3 | 60.7 |
| **+ TrackIME** | **57.9** | **72.4** | **66.9** | **80.0** | **51.0** | **63.6** |
| PIPS2 [9] | 46.6 | 69.4 | 52.3 | 74.9 | - | - |
| **+ TrackIME** | **50.3** | **74.0** | **52.8** | **75.8** | - | - |
| CoTracker [7] | 60.8 | 76.1 | 64.1 | 78.0 | 47.7 | 63.7 |
| **+ TrackIME** | **64.5** | **79.2** | **68.2** | **82.1** | **48.1** | **63.8** |
| OmniMotion† [8] | 52.6 | 68.1 | 71.2 | 81.1 | 51.0 | 64.3 |
| **+ TrackIME** | **54.1** | **69.3** | **71.9** | **81.9** | **51.2** | **64.6** |
| TAPIR [6] | 57.5 | 70.5 | 66.3 | 80.6 | 50.2 | 62.3 |
| **+ TrackIME** | **65.3** | **78.6** | **66.6** | **81.8** | **51.4** | **65.8** |

In Section 4.2, we verify the efficacy of the enhanced point tracking results by TrackIME in the downstream video object segmentation. Specifically, we compare the zero-shot video segmentation performances with the recent SAM-PT [30] baseline which utilizes the point trajectories as the inputs, as well as the conventional baselines that input the semantic classes [27, 28, 29].

**Common implementation details.** We note that TrackIME is mainly incorporated with TAPIR point tracker [6] (as it empirically performs best) unless specified otherwise, and we subject it to all experiments including the point tracking and other downstream tasks.

For the segmentation model, we utilize the Segment Anything (SAM) [1] to perform the point-queried segmentation function described in Equation (5).

To prepare video frames, we always adjust the resolutions of raw video data to 1080p (1080 pixels in the shorter frame edges), then apply further resizing functions required by individual baseline models. For example, we resize the 1080p frames to $256 \times 256$ for TAPIR [6] baseline, following the default setting provided by the official open-source repository. When experimenting TrackIME, we choose the hyperparameters for each baseline, $e.g.$, progressive inference steps $K = 2$, and the pruning sizes $H_0 = W_0 = 960$ and $H_1 = W_1 = 384$ when incorporated with TAPIR [6].

Since TrackIME is a plug-in to all baselines, we reproduce all results in the same system configuration for fair comparisons. We note that such modification can induce minor perturbation in the numerical values due to library and hardware-dependent characteristics, $e.g.$, different characteristics between JAX [31] and PyTorch [32] libraries, and the difference in the filtering algorithm used when re-sizing the video frames.[5] We refer the readers to Appendix A for more implementation details.

## 4.1 Point Tracking

**Baselines.** We compare our method to the recent baselines OmniMotion[8], CoTracker [7], TAPIR [6], PIPS2 [9], and TAPNet [11]. We utilize the official checkpoints provided by the official project pages and reproduce all experimental results under our common experimental set-up, except for OmniMotion [8] which does not provide checkpoints. Instead, we reproduced the training of OmniMotion models to obtain the experimental results. We use $S = 31$ semantic neighbors to incorporate our framework with the baselines.

**Datasets.** We evaluate these models on three different datasets, DAVIS [12], Kinetics [34], and RGB-Stacking [33], each representing different characteristics. For example, DAVIS contains 30 videos

---

[5]The open-source version of TrackIME is available at https://github.com/kami93/trackime.

Table 3: **Ablation study of the components in our model.** We ablate the effect of search space pruning (Pruning), trajectory aggregation (Aggregation), and the progressive inference (Progressive) modules for point tracking. We evaluate the tracking benchmark in DAVIS scenes [11, 12].

| Pruning | Aggregation | Progressive | $J_1$ | AJ | $\delta_1^x$ | $\delta_{avg}^x$ |
|---------|-------------|-------------|-------|------|--------------|------------------|
| ✗ | ✗ | ✗ | 23.0 | 57.5 | 34.3 | 70.5 |
| ✓ | ✗ | ✗ | 28.2 | 62.5 | 41.1 | 75.3 |
| ✓ | ✗ | ✓ | 28.3 | 62.6 | 41.2 | 75.6 |
| ✓ | ✓ | ✗ | 34.0 | 62.9 | 48.0 | 77.0 |
| ✓ | ✓ | ✓ | **35.4** | **65.3** | **48.2** | **78.6** |

specifically curated to evaluate the tracking performance under large variances in the appearance and motion of object entities. Its two variants, DAVIS-F (First) and DAVIS-S (Strided) differ in how the query points are given to the models: DAVIS-F queries the model only once in the first frame, while DAVIS-S queries the model in strides of five frames. Because DAVIS-F requires long-term tracking, it is generally a more difficult setting. Kinetics contains 1,144 web videos collected from YouTube that represent realistic noisy characteristics of the video in the wild, such as sudden scene changes. RGB Stacking is a synthetically rendered dataset representing 50 different moves by a robotic arm. For all datasets, we refer to the point tracking annotations provided by TAP-Vid [6] and utilize them as the ground truth for evaluation.

**Metric.** To measure the quality of point tracking, we consider point tracking accuracy considered following TAP-Vid [11], such as the $\delta$-average accuracy ($\delta_{avg}^x$) and the average Jaccard (AJ). The average metrics are based on the $\delta$-n accuracy ($\delta_n^x$) which indicates the proportion of correct trajectory sequence as judged by whether they are within the n-pixel error threshold around the ground truth. In addition, the Jaccard-n ($J_n$) judges a trajectory sequence to be correct only if the visibility prediction is also correct. Given these definitions, the average metrics are calculated by averaging $n \in \{1, 2, 4, 8, 16\}$. To evaluate the fine-grained tracking performance in a harsh error threshold, we also report $\delta$-1 accuracy ($\delta_1^x$) and Jaccard-1 ($J_1$). For Table 1, we also discuss the occlusion accuracy (OA), the proportion of correct visibility sequence given the ground truth.

**Effectiveness on point tracking in dynamic objects.** We first present the point tracking scenarios with dynamic objects. Specifically, we experiment with the DAVIS video scenes [12], which is curated for evaluating instance motion estimation tasks. As shown in Table 1, we find our method achieves the best point tracking accuracy surpassing all baselines, *e.g.*, up-to 7.4% relative improvements in average Jaccard, *i.e.*, 60.8 AJ (CoTracker [7]) vs. 65.3 AJ (TrackIME) when evaluated with the DAVIS-F (denoted First Query in Table 1). We also measure the occlusion accuracies (OA) and find a relatively incremental improvement than other metrics. Intuitively, there is a trade-off between modeling the occlusions among different objects and the search space pruning for one instance, as the pruning removes information from other instances. Nevertheless, our method is beneficial for detecting occlusion in fine-grained object parts, and we recommend searching for optimal pruning parameters that fit a user's purpose. Finally, we discuss the efficacy of TrackIME under the harsh $\delta_1^x$ and $J_1$ metrics, where the conventional metrics allows up to 16-pixel errors and takes the average when judging whether the prediction is correct. For example, the improvement can be even larger, *e.g.*, up to relative 25.1%, *i.e.*, 28.3 $J_1$ (CoTracker [7]) vs. 35.4 $J_1$ (TrackIME) when evaluated with DAVIS-F. We highlight these benefits of TrackIME allowed by pruning the search space.

**Universality to different point tracking models.** We validate the universality of our method when plugged into the state-of-the-art baselines by evaluating the average tracking accuracy (AJ and $\delta_n^x$) of the vanilla models and the variants incorporated with our method in Table 2 on DAVIS (First) [12], RGBStacking [33], and Kinetics [34] datasets. As a result, we observe that our method can provide consistent and significant performance improvements in all the baselines, *e.g.*, 13.6% relative improvements (*i.e.*, 57.5 → 65.3 AJ) in TAPIR [6] when evaluated on the DAVIS. Since the model variant incorporated with TAPIR demonstrates the best performance, we chose it as our main model and subjected it to other studies. We note that the experiments for OmniMotion [8] have been conducted in 16 subsets for RGBStacking and Kinetics, and $K = 1$ progressive inference, due to its

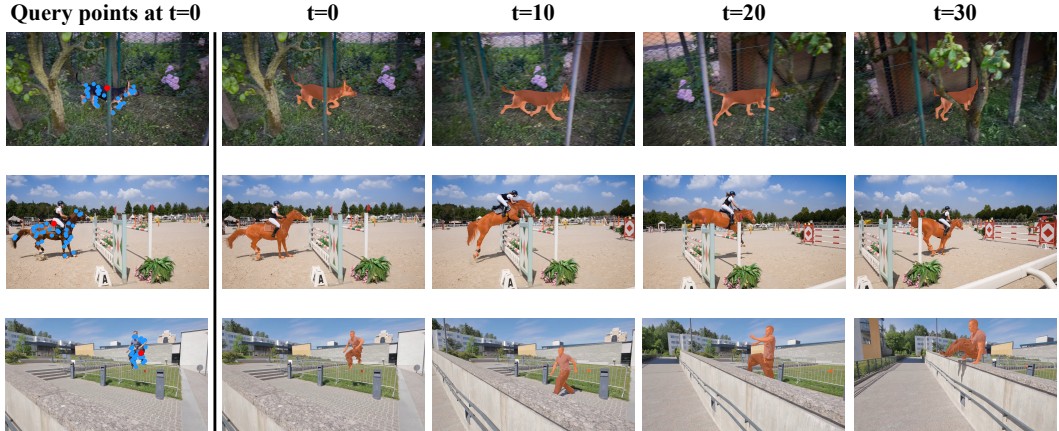

| Query points at t=0 | t=0 | t=10 | t=20 | t=30 |

Figure 2: **Demonstration of the video instance segmentation results by our TrackIME framework.** Given the query points in the reference frame, our framework can produce the video instance segmentation masks at quality by performing the weighted aggregation of the mask associated each query point, based on the visibility values.

heavy optimization costs, *e.g.*, approximately 13 gpu-hours for processing one scene. We also note that PIPS2 [9] in Kinetics [12] is unavailable, as its memory requirement for processing Kinetics exceeds our system's capacity.

**Ablation study.** We perform an ablation study to understand how each component affects the point trajectory accuracy in Table 3. Specifically, we consider the search space pruning, the trajectory aggregation, and progressive inference modules as the subjects for the ablation.

First of all, we reveal the pure efficacy of our pruning method, separate from the effect of segmentation prior. Notably, when the trajectory aggregation module is removed (the first 2 rows in Table 3), we observe the pruning solely based on the query point's trajectory provides the most significant effect (*e.g.*, $23.0 \rightarrow 28.2$ in $J_1$). This validates our key motivation for pruning the search space, which provides superior results even if SAM [1] is not employed.

Next, we discuss the effect of employing SAM [1] by enabling the trajectory aggregation. As expected, aggregating the trajectories for a group of points found in the segmentation mask provides another comparable gain (*e.g.*, $28.2 \rightarrow 34.0$ in $J_1$), which validates that the aggregation improves the quality of instance trajectory estimation.

It is worth noting that the progressive inference boosts the performance, (*e.g.*, $34.0 \rightarrow 35.4$ in $J_1$) when combined with the trajectory aggregation, otherwise the gain is lesser (*e.g.*, $28.2 \rightarrow 28.3$ in $J_1$). As the progressive inference refers to the estimated instance trajectory, the estimation quality is essential for this module.

We also note that further ablation study is available in Appendix D, *e.g.*, the number of semantic neighbors, progressive inference steps, or the pruning sizes.

## 4.2 Video Object Segmentation

In this section, we validate the efficacy of TrackIME by performing the zero-shot video segmentation. We also provide the visualization results for selected scenes from DAVIS [12] in Figure 2.

**Baselines.** We experiment with zero-shot video object segmentation to check the efficacy of TrackIME for improving segmentation. Specifically, we consider the class-guided baselines for unsupervised video segmentation tasks, *e.g.*, EntitySeg [29]. In addition, we consider the SAM-PT [30] baseline which also proposes to take point tracking for producing segmentation. To consider the equivalent experimental set-ups for SAM-PT [30] and TrackIME, we incorporate the models with

Table 4: **Zero-shot video object segmentation performance in DAVIS benchmark.** We consider two set of zero-shot baselines, those utilizing the set of classes [23, 24, 25, 26, 27, 28, 29] and the baseline utilizing a set of query points [30] in a similar manner to our TrackIME. †: we produced the results for TrackIME and SAM-PT [30] under the common set-up, such as the number of tracking points, segmentation function (HQ-SAM [35]), and the same mask formatting for the benchmark.

| Method | Input | DAVIS-2017-val | | | DAVIS-2017-test-dev | | |
| --- | --- | --- | --- | --- | --- | --- | --- |
| | | $(J\&F)_m$ | $J_m$ | $F_m$ | $(J\&F)_m$ | $J_m$ | $F_m$ |
| PDB [23] | class | 55.1 | 53.2 | 57.0 | 40.4 | 37.7 | 43.0 |
| RVOS [24] | class | 41.2 | 36.8 | 45.7 | 22.5 | 17.7 | 27.3 |
| AGS [25] | class | 57.5 | 55.5 | 59.5 | 45.6 | 42.1 | 49.0 |
| MAST [26] | class | 65.5 | 63.3 | 67.6 | - | - | - |
| Propose-Reduce [27] | class | 70.4 | 67.0 | 73.8 | - | - | - |
| UnOVSOT [28] | class | 67.9 | 66.4 | 69.3 | 58.0 | 54.0 | 62.0 |
| EntitySeg [29] | class | 73.4 | 70.4 | 76.4 | 62.1 | - | - |
| SAM-PT† [30] | points | 78.8 | 76.3 | 81.3 | 65.3 | 62.3 | 68.3 |
| **TrackIME†** | points | **79.6** | **76.4** | **82.8** | **65.9** | **62.5** | **69.4** |

HQ-SAM [35] variant for the segmentation, 16 points from the initial frame's mask, and employ the iterative refinement technique [35] to produce the video segmentation results.

**Evaluation.** We evaluate our model on the DAVIS-2017 [12] video segmentation. In particular, we use the validation and the test-dev sets for the zero-shot benchmark. Both sets contain 30 non-overlapping scenes with single or multiple objects.

To measure the quality of video instance segmentation, we consider the standard metrics in baselines: the mean Jaccard ($J_m$); the mean F-measure ($F_m$); and the average $(J\&F)_m$. Specifically, we follow the official implementation suite provided by the DAVIS challenge [12].

**Effectiveness on zero-shot video object segmentation.** In Table 4, we first confirm that the point tracking provides useful guidance for video segmentation, observing that both SAM-PT [30] and TrackIME demonstrates significant improvement over the conventional class-prompted baselines. More importantly, as our framework brings synergistic improvements for both point tracking and segmentation tasks, we find TrackIME achieves even larger improvement, *e.g.*, 78.8 vs. 79.6 $(J\&F)_m$ in the validation set of DAVIS-2017 [12].

**Discussions.** As for the commentary on the efficacy of TrackIME, our key advantage is removing erroneous query points for segmentation caused by the tracking failure on intricate object parts, enabling even finer query points for segmentation, *e.g.*, the accuracy in harsh 1-pixel thresholds in Table 1, which is possible due to the pruning structure in our framework to maintain the high-frequency information.

## 5 Conclusion

In this work, we introduce TrackIME, a novel approach for point tracking to overcome the fundamental challenge of computation demands in existing models. Specifically, we reduce the search space by identifying the instance trajectory and pruning the video frames along it. To obtain the instance trajectory, we aggregate the motion for a group of points on the segmentation masks. To this end, we propose a unified framework that jointly performs point tracking and segmentation, with the techniques to ensure robustness to occlusion in complex video scenes. TrackIME demonstrates consistent and significant impacts by bolstering existing point tracking baselines. The joint framework also reveals the synergistic effects, which also demonstrates the improvements in the video segmentation task. Overall, our work highlights the effectiveness of considering instance motion trajectory and jointly solving the tracking and segmentation, and we believe our work could inspire researchers to consider a new direction to further leverage it in the future.

## Acknowledgements

This work was partly supported by Institute of Information & communications Technology Planning & Evaluation (IITP) grant funded by the Korea government (MSIT) (No.RS-2019-II190075, Artificial Intelligence Graduate School Program(KAIST); No.RS-2021-II212068, Artificial Intelligence Innovation Hub; No.RS-2020-II201819, ICT Creative Consilience Program), and Culture, Sports and Tourism R&D Program through the Korea Creative Content Agency grant funded by the Ministry of Culture, Sports and Tourism in 2024(Project Name: International Collaborative Research and Global Talent Development for the Development of Copyright Management and Protection Technologies for Generative AI, Project Number: RS-2024-00345025).

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

# Appendix

## A Experimental details for point tracking

In this section, we present detailed experimental setups considered by our experiments in Section 4.

**Baselines.** We consider 5 different baseline point tracking models, TAPNet [11], PIPS2 [9], CoTracker [7], OmniMotion [8], and TAPIR [6]. We experiment with the checkpoint provided in the official open-source repository hosted by their authors, following the default hyperparameters in each model, *e.g.*, for the input dimensions, in TAPNet [11] and TAPIR [6] consider a square-shaped ($256 \times 256$) dimension, while PIPS2 and CoTracker do rectangular-shaped dimensions, ($512 \times 896$) and ($384 \times 512$), respectively. We also note that the backend library of TAPIR and TAPNet is ported from JAX [31] to PyTorch [32] in our experiments, which provides subtle enhancements in the tracking accuracy, *e.g.*, AJ 56.2 [6] $\rightarrow$ 57.5 (Table 2) in DAVIS-F.

Table 5: **The pruned resolutions in our method for each baseline point tracking model.** We report the specific values for $H_0, H_1, W_0, W_1$ when TAPNet[11], PIPS2[9], CoTracker[7], OmniMotion [8], or TAPIR[6] is used as the baseline.

| Baseline Model | $H_0$ | $H_1$ | $W_0$ | $W_1$ |
|---|---|---|---|---|
| TAPNet [11] | 960 | 384 | 960 | 384 |
| PIPS2 [9] | 960 | 512 | 1680 | 896 |
| CoTracker [7] | 960 | 384 | 1280 | 512 |
| OmniMotion [8] | 960 | – | 960 | – |
| TAPIR [6] | 960 | 384 | 960 | 384 |

**Hyperparameters for point tracking.** Unless otherwise specified, we always choose the number of semantic neighbors $S = 31$, and the progressive inference steps $K = 2$ for TAPNet [11], PIPS2 [9], CoTracker [7], and TAPIR [6]. For OmniMotion [8], we set the progressive inference step $K = 1$. To incorporate our framework with the baselines, we select different pruning sizes to meet the shape requirements of a specific model (*e.g.*, TAPIR [6] needs a shape in multiples of 8 to be compatible with its convolution layers). For example, if our method is plugged into TAPIR [6] and a video with the 1080p resolution, *e.g.*, ($1080 \times 1920$), we set the pruning resolutions $H_0 = W_0 = 960$ and $H_1 = W_1 = 384$. For clarity, we present the resolutions for all baseline models in Table 5.

**Datasets.** We evaluate the baselines and TrackIME in three different datasets from the TAP-Vid benchmark [11]: DAVIS [12]; Kinetics [34]; and RGBStacking [8]. The sizes of the raw samples can vary, *e.g.*, from 256 to 2160 in their shorter sides, hence we process the frames by resizing the shorter sides to 1080 with the aspect ratio fixed. As a result, the video frame resolutions are typically ($1080 \times 1920$) for DAVIS [12] and Kinetics [34]. We note that RGB-Stacking is originally in ($256 \times 256$), but we do bilinear up-sampling to ($1080 \times 1080$) for simplicity.

**Experimental environment.** Every baseline model and internal module in TrackIME (*e.g.*, Segment Anything [1]) is implemented in PyTorch 2.1 [32] compiled for CUDA 11.8, which we run on an NVIDIA RTX 3090 GPU. In default, we experiment with the float32 numerical precision; however, in case of out-of-memory errors (*e.g.*, RTX 3090's 24 GiB VRAM cannot handle hundreds of frames), we employ the bfloat16 precision to fit such samples into the limited memory.

## B Backgrounds

In this section, we describe technical details behind the limitations in the current point tracking models.

In the common canonical design of recent model architectures for Equation (1), *e.g.*, our baselines: TAPNet [11], CoTracker [7], TAPIR [6], etc., the key component is the cost volume [21], which represents the likelihood of the query point's spatial-temporal location over the entire video frames. In principle, predicting this cost map requires a brute-force search over every spatial-temporal location, which is often computationally infeasible on the raw video dimensions, *e.g.*, 1080p. To mitigate this problem, current models first down-sample the raw video into a lower spatial resolutions,

Table 6: **FLOP counts by each module in TrackIME.** We report the FLOP counts for point tracking given 64 video frames, during the instance motion stage, and the high-fidelity tracking with $K = 2$ progressive steps.

| TrackIME Modules | FLOPs |
|---|---|
| **Instance trajectory estimation** (stage 1) | **1355G** |
| - Segmentation | 533G |
| - Instance Tracking (32 points; $S = 31$) | 822G |
| **Progressive inference** (stage 2) | **1434G** |
| - $k = 0$ (32 points; $S = 31$) | 822G |
| - $k = 1$ (1 point) | 612G |
| **Total** | **2789G** |

*e.g.*, ($256 \times 256$) in TAPIR [6]. While the reduced resolution enables models to process the entire video frames for tracking, the lost information during the resolution reduction induces quantization noises into the cost volume. Recent baselines, including the state-of-the-art [6], employ refinement techniques to mitigate these noises.[6] Nevertheless, the lost detail in the visual feature after the down-sampling still hinder representing high-frequency patterns, and the model can suffer from tracking failure modes.

In this regard, our method pursues the direction of pruning the excessive search space for point tracking, so that models can avoid the down-sampling and focus only on important regions maintaining detailed visual features.

## C  Computational costs for point tracking

In this section, we study the computational costs and efficiency of TrackIME by examining the FLOP (floating-point operations) counts for performing the point tracking.

**FLOP count of TrackIME.** To check the exact cost of each module in TrackIME, we report the FLOPs for tracking under our default setting, *e.g.*, TAPIR [6] as the baseline, given 64 video frames. Specifically, as given in Table 6, the segmentation with SAM [1] needs 533 GFLOPs, tracking 32 points (*e.g.*, $S = 31$ semantic neighbors plus one query point) demand 822 GFLOPs, and tracking a single point demands 612 GFLOPs, respectively. As a result, the net FLOP count of TrackIME (with $K = 2$ progressive steps) is 2789 GFLOPs.

**Computation efficiency compared to baselines.** Next, we compare the baseline TAPIR [6] with various input dimensions and TrackIME, in terms of their FLOP counts versus the point tracking performances, AJ (Average Jaccard), $\delta_{\text{avg}}^x$, and OA (Occlusion Accuracy), evaluated under DAVIS-F and DAVIS-S in Table 7.

For TAPIR, the FLOP count is mostly governed by the input dimension of a model ($256 \times 256$), *e.g.*, 612 GFLOPs for processing 64 video frames, and it grows quadratically as the input dimension gets increased.

An interesting finding in Table 7 is that the baseline [6] cannot benefit from the larger input dimensions without fine-tuning. For example, we observe that the baseline's performance only deteriorates given larger inputs, as the model is only optimized for a low-resolution input frames ($256 \times 256$) to meet the memory constraints while training; it is non-trivial to process high-resolution inputs without fine-tuning. Furthermore, even if fine-tuning is employed (*e.g.*, TAPIR Hi-Res [6]), the performance gain (*e.g.*, $62.8 \rightarrow 65.7$ AJ) is not significant considering the excessive increase in FLOP counts (*e.g.*, $612 \rightarrow 8257$ GFLOPs), and the occlusion accuracy (OA) can even get worse (*e.g.*, $88.3 \rightarrow 86.7$).

These results further demonstrate the merits of employing TrackIME for point tracking, which can enable point tracking models to process the frames in a computationally efficient manner, even without fine-tuning, and provide consistent performance gains. For example, comparing TrackIME (ours) vs.

---

[6]We refer the readers to literature for the refinement mechanisms [6, 7].

Table 7: **The comparison of the FLOP counts of the TAPIR [6] models and TrackIME.** We report the FLOP counts to process 64 video frames by TAPIR with the input dimensions $(256 \times 256)$ (default), $(512 \times 512)$, and $(768 \times 768)$, TAPIR Hi-Res (a fine-tuned model for $(1080 \times 1080)$) and TrackIME (ours). For each model, we further report the benchmark results in terms of AJ (Average Jaccard), $\delta_{\text{avg}}^x$, and OA (Occlusion Accuracy), evaluated under DAVIS-F and DAVIS-S. For TAPIR Hi-Res, numbers are excerpted from [6], where results for DAVIS-F are not available.

| Method (Input Dim.) | FLOPs | DAVIS-F | | | DAVIS-S | | |
|---|---|---|---|---|---|---|---|
| | | AJ | $\delta_{\text{avg}}^x$ | OA | AJ | $\delta_{\text{avg}}^x$ | OA |
| TAPIR $(256 \times 256)$ | 612G | 57.5 | 70.5 | 85.5 | 62.8 | 75.1 | 88.3 |
| TAPIR $(512 \times 512)$ | 2429G | 53.9 | 65.9 | 79.8 | 62.5 | 74.0 | 81.8 |
| TAPIR $(768 \times 768)$ | 5457G | 53.3 | 65.5 | 73.2 | 58.3 | 70.2 | 76.6 |
| TAPIR Hi-Res $(1080 \times 1080)$ | 8257G | - | - | - | 65.7 | 77.6 | 86.7 |
| **TrackIME** $(256 \times 256)$ | 2789G | **65.3** | **78.6** | **86.5** | **69.3** | **81.4** | **89.0** |

Table 8: **The comparison of the FLOP counts of the CoTracker [7] models and TrackIME.** We report the FLOP counts to process 64 video frames by CoTracker with the input dimensions $(384 \times 512)$ (default), $(768 \times 1024)$, and $(1080 \times 1440)$ and TrackIME (ours). For each model, we further report the benchmark results in terms of AJ (Average Jaccard), $\delta_{\text{avg}}^x$, and OA (Occlusion Accuracy), evaluated under DAVIS-F.

| Method (Input Dim.) | FLOPs | DAVIS-F | | |
|---|---|---|---|---|
| | | AJ | $\delta_{\text{avg}}^x$ | OA |
| CoTracker $(256 \times 256)$ | 2707G | 60.8 | 76.1 | 86.0 |
| CoTracker $(512 \times 512)$ | 7670G | 62.3 | 77.8 | 87.1 |
| CoTracker $(768 \times 768)$ | 5457G | 62.2 | 76.7 | 86.6 |
| **TrackIME** $(384 \times 512)$ | 6217G | **64.5** | **79.2** | **88.5** |

TAPIR Hi-Res [6] gives: **2789G** vs. 8257G (FLOPs); **69.3** vs. 65.7 (AJ); **81.4** vs. 77.6 ($\delta_{\text{avg}}^x$); and **89.0** vs. 86.7 (OA), in DAVIS-S, respectively.

In the similar manner, we also provide the FLOPs count for our method incorporated with [7] in Table 8.

# D  Ablation study

In this section, we ablate the choice of hyperparameters in our enhanced point tracking, namely the pruning sizes $(H_0, W_0)$ without the progressive fusion (*i.e.*, $K = 1$) and our default setting in TrackIME ($K = 2$), and the number of sampling semantic neighbors $S$ for estimating the instance trajectory.

In Table 9, we find that smaller pruning sizes tend to introduce positive effects in the fine-grained metrics (*e.g.*, 1- and 2-pixel error thresholds), but also trade off the average-scale metrics (*e.g.*, AJ and $\delta_{\text{avg}}^x$). These results are expected, as the pruning size gets smaller, the amount of down-sampling reduces and more detailed visual features would be preserved, but at the same time, the chance of erroneous pruning increases where the true location of the query point is lost.

Table 9: **Ablation study of the pruning size in our framework.** We ablate the pruning size considered in TrackIME. For the evaluation, we calculate both pixel-scale and average-scale metrics under the DAVIS-F dataset [11].

| Pruning Size ($K$) | $J_1$ | $\delta_1^x$ | $J_2$ | $\delta_2^x$ | AJ | $\delta_{\text{avg}}^x$ |
|---|---|---|---|---|---|---|
| 1080 ($K=1$) | 28.1 | 41.0 | 52.3 | 66.0 | 62.5 | 75.2 |
| 960 ($K=1$) | 29.7 | 42.4 | 52.9 | 66.3 | 63.1 | 75.6 |
| 768 ($K=1$) | 31.9 | 44.6 | 55.7 | 68.1 | 64.0 | 76.4 |
| 512 ($K=1$) | 34.6 | 47.6 | 56.5 | 68.9 | 63.9 | 76.9 |
| 384 ($K=1$) | 35.3 | 47.3 | 56.5 | 68.3 | 62.3 | 76.1 |
| $960 \rightarrow 384$ ($K=2$) | **35.4** | **48.2** | **57.7** | **70.1** | **65.3** | **78.6** |

Table 10: **Ablation study of the effect of the number of semantic neighbors in our method.** We ablate the number of semantic neighbors considered in our method. For the evaluation, we calculate both pixel-scale and average-scale metrics under the DAVIS-F dataset [11].

| $S+1$ | $J_1$ | $\delta_1^x$ | $J_2$ | $\delta_2^x$ | AJ | $\delta_{\text{avg}}^x$ |
|---|---|---|---|---|---|---|
| 128 | 35.1 | 47.7 | 57.1 | 69.4 | 64.8 | 78.2 |
| 64 | 35.0 | 47.5 | 57.2 | 69.8 | 64.8 | 78.3 |
| 32 | **35.4** | **48.2** | **57.7** | 70.1 | **65.3** | **78.6** |
| 16 | 35.2 | 47.9 | 57.3 | 69.9 | 64.8 | 78.5 |
| 8 | 35.1 | 47.8 | 57.4 | **70.2** | 64.9 | 78.5 |
| 4 | 35.0 | 47.6 | 57.5 | 69.8 | 64.9 | 78.3 |
| 2 | 35.1 | 47.9 | 57.2 | 69.7 | 64.7 | 78.1 |

We note that the progressive fusion ($K=2$) in our method can mitigate the trade-off in pruning by considering multiple scales, *e.g.*, $H_0 = 960$ and $H_1 = 384$, providing additional performance gains.

Next, in Table 10, we ablate the effect of the choice for the number of semantic neighbors $S+1$ (including the query point), halving down its value starting from $(S+1) = 128$ to $(S+1) = 2$. As a result, we find that all of the choices can provide satisfactory performance in general, although there exist mild trade-offs between the 1- and 2-pixel scale metrics and the average scale metrics. As one of our goal is on achieving the optimal pixel-scale performance in point tracking, we empirically choose $(S+1) = 32$, which reveals the best 1-pixel scale metrics.

## E    Additional experiments and visualizations

In this section, we provide the additional experiment and visualizations with TrackIME.

**The use of visibilities as the confidence weights.** Our strategy combines both the hard 0-1 visibility predictions as well as the confidence weights (*e.g.*, Equation (3)). This strategy effectively mitigates potentially erroneous confidences by the false positives, since our method tends to demonstrate a high precision (the portion of true positives) for the visibility classification, e.g., we get 93.7% at the threshold 0.5 in DAVIS-F. Our strategy is valid as far as a sufficient number of visible tracking points are available. For the cases where an object is occluded for a few frames and then reappears, our framework can maintain the number of tracking points via the point re-sampling, even if the visibility classifier fails to predict the reappearance.

To further support the validity of our strategy, we measure the average confidence of the true positives and the false positives and find 0.902 (true positives) and 0.737 (false positives), so the remaining false positives would be penalized through the weighted aggregation. We also provide additional study in Table 11, where we force equal weights in the aggregation experimented in DAVIS-F. For example, we find 1.3 points improvement by using our strategy.

**TAPNet results with an alternative checkpoint.** In our main experiments, we have utilized ResNet18 backbone image backbone provided by the official checkpoint to reproduce TAPNet [11] and TAPIR [6] results, instead of TSM-ResNet18 used by TAPNet in the original paper [11]. In Table 12, we

Table 11: **Forcing equal weights in the aggregation**. We ablate the use of aggregation weights in our method. For the evaluation, we calculate both pixel-scale and average scale metrics under DAVIS-F dataset [11].

| Method | $J_1$ | AJ | $\delta_1^x$ | $\delta_{\text{avg}}^x$ | OA |
|---|---|---|---|---|---|
| TrackIME (equal weights) | 34.9 | 64.0 | 47.9 | 78.5 | 86.3 |
| TrackIME (default) | **35.4** | **65.3** | **48.2** | **78.6** | **86.5** |

additionally provide the results based on the checkpoint with the original TSM-ResNet18 image backbone. When experimented with DAVIS-F and DAVIS-S, we find TrackIME keeps demonstrating significant gains, *e.g.*, $32.8 \rightarrow 47.0$ AJs (14.2 points) in DAVIS-F.

Table 12: **TAPNet results with an alternative checkpoint**. We experiment with the use original TSM-ResNet18 image backbone for the TAPNet baseline [11]. For the evaluation, we calculate both pixel-scale and average scale metrics under DAVIS-F and DAVIS-S datasets [11].

| Method | First Query | | | | | Strided Query | | | | |
|---|---|---|---|---|---|---|---|---|---|---|
| | $J_1$ | AJ | $\delta_1^x$ | $\delta_{\text{avg}}^x$ | OA | $J_1$ | AJ | $\delta_1^x$ | $\delta_{\text{avg}}^x$ | OA |
| TAPNet [11] | 5.8 | 32.8 | 11.1 | 48.4 | 77.6 | 6.7 | 38.4 | 12.6 | 53.4 | 81.4 |
| + **TrackIME** | **18.7** | **47.0** | **28.6** | **60.6** | **80.9** | **21.5** | **50.8** | **32.4** | **63.8** | **81.4** |

**Visualization of the progressive inference.** We additionally visualize the progressive inference structure in Figure 3. Specifically, we incorporated TrackIME with TAPIR [6] and apply the progressive pruning sizes of $(960 \times 960)$ and $(384 \times 384)$. As depicted by Figure 3, the latest progressive step is well focused around the query point, *e.g.*, the dog's ear, so that the search space for point tracking is effectively pruned.

# F    Limitation

## F.1    Limitation and Future Works

TrackIME relies on the pre-trained models for point tracking, often trained with synthetic datasets, such as Kubric [36] and PointOdyssey [9], while the segmentation models are primarily trained on the real images [1]. An interesting future direction is to integrate the TrackIME with training on the real video scenes. As this could include the development of point tracking algorithms capable of generalization to diverse intricate objects or, alternatively, optimizing the segmentation models for better video scene understanding. This approach could further improve the accuracy and applicability of TrackIME in various real-world scenarios.

## F.2    Potential Negative Societal Impact

While point tracking by TrackIME can be beneficial for various video understanding applications, such as novel-view synthesis, depth estimation, and action recognition, the emergence of unexpected behavior within TrackIME can lead to misrepresentations of the real video data. For those applications that require extremely accurate models for safety-related judgements, such as depth estimation for autonomous driving, the unexpected behaviors must be carefully managed. To ensure the reliability of systems using point tracking predictions, we recommend to conduct thorough investigations and implement robust mitigation strategies to minimize potential risks, thereby increasing the overall safety and effectiveness of these applications.

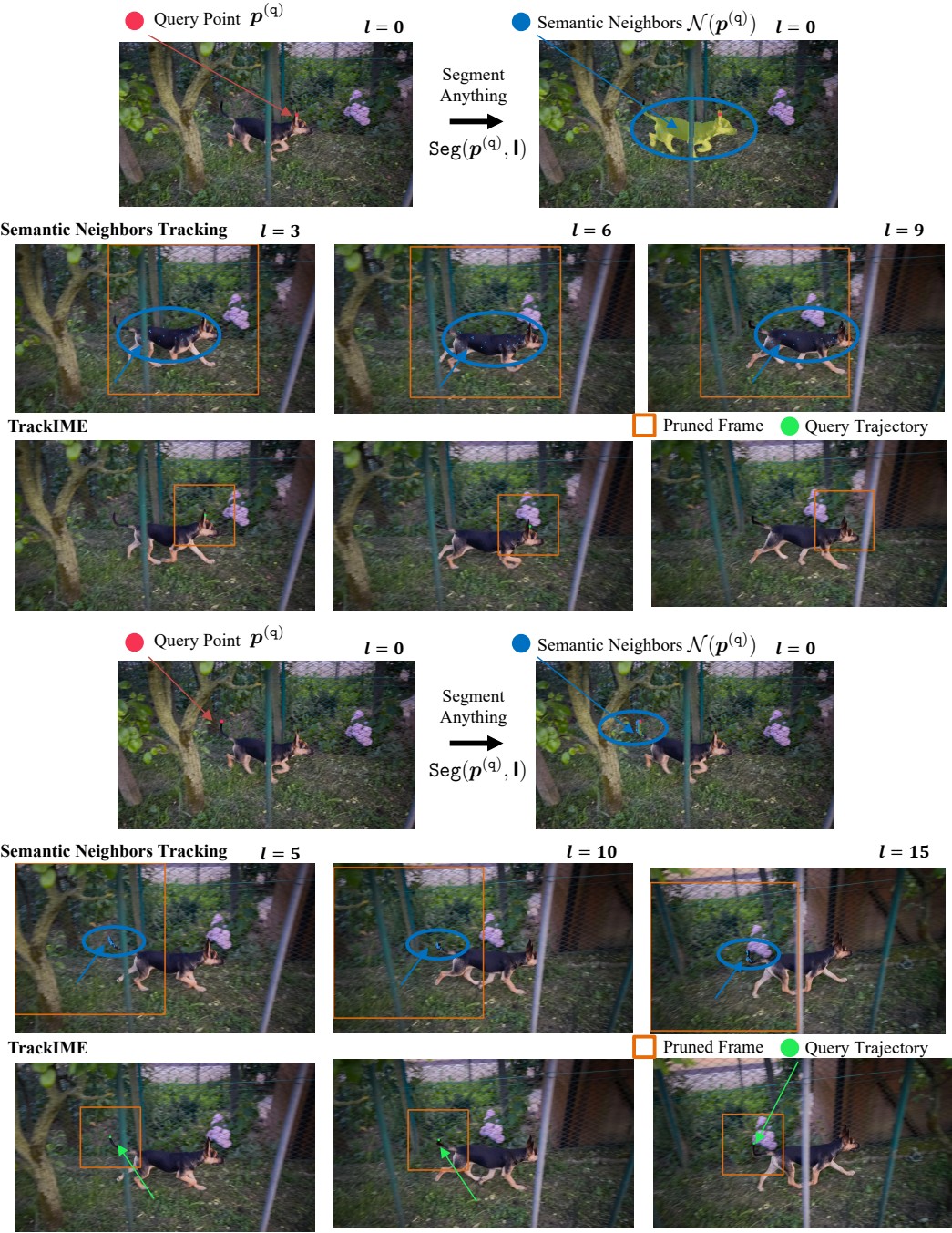

Figure 3: **Demonstration of the progressive inference by TrackIME framework.**

