# OpenReview forum: "TrackIME: Enhanced Video Point Tracking via Instance Motion Estimation"
_NeurIPS.cc/2024/Conference — NeurIPS 2024 spotlight_

### Official Review · Reviewer_7a51 · 2024-07-07

**Soundness:** 3
**Presentation:** 3
**Contribution:** 2
**Rating:** 5
**Confidence:** 3

**Summary:**

In this paper,  the authors propose TrackIME, a framework to tracking points in video. The proposed TrackIME leverages a segmentation model to improve both its efficient and effectiveness. TrackIME achieves SOTA performance on TAP-VID dataset.

**Strengths:**

1. The proposed method achieves competitive results while retaining a low computation cost (comparing with TAPIR)
2. Using segmentation model to help points tracking has never been explored by prior works.

**Weaknesses:**

1. The main contribution of proposed method is not clear to me. Using segmentation model to help track points within an instance is an intuitive top-down solution for points tracking
2. The proposed method outperform SAM-PT in a relative small margin in terms of instance-level tracking, which raise my concerns about the capability of model for tracking points. It seems to me that the improvement on the point tracking performance is limited to help tracking instances/objects.

**Questions:**

1. The authors only compare FLOPs between TrackIME and TAPIR, why not also compare with other methods like CoTracker?

---

> ### Author Rebuttal · Authors · 2024-08-07
>
> Dear reviewer 7a51,
>
> Thank you for your valuable feedback and comments. We appreciate your remarks on the strengths of our paper, including the competitive results, low computation cost, and the first introduction of the segmentation model in point tracking tasks. We will address your concerns and questions in the response below.
>
> ---
>
> **[W1] The main contribution of our paper**
>
> Thank you for acknowledging our intuitive idea of using segmentation models to enhance point tracking. We list our main contributions, which are commonly acknowledged by the reviewers jQxW and BSTo, as follows:
>
> - Designing a flexible model that can combine different pre-trained tracker and pre-trained instance segmentation models, not tied to a specific model
> - A new idea of pruning input video frames for point tracking, which addresses a fundamental drawback in existing point tracking models caused by the video down-sampling structure
> - A joint framework for the point tracking and the segmentation tasks, which bolsters the performance of the two tasks in a synergistic manner
>
> As pointed out by the reviewer jQxW, our method addresses the fundamental limitation of losing high-frequency information crucial for point tracking, innovating the way current point tracking models handle high-dimensional video inputs.
>
> As the segmentation improves point tracking in our framework, it can also enhance the segmentation task in a synergistic manner. Specifically, the enhanced fine-grained tracking performance (e.g., 1-pixel errors in Table 1) can bolster the segmentation task by providing more accurate queries for segmentation. While prior art (e.g., SAM-PT) contributed to enhancing the segmentation from the point tracking task, there has been no recursive structure similar to ours, and their performance gain is only a one-off.
>
> Finally, we note that our method does not require fine-tuning, and is pluggable to different existing models (Table 2). It is also worth noting that the efficacy of our pruning method is significant even if the segmentation model is not employed (e.g., 57.5 -> 62.5 AJ in Table 3).
>
> ---
>
> **[W2] Comparison to SAM-PT**
>
> Despite the margin being relatively small in the segmentation task, we note that our primary focus is point tracking where our framework demonstrates significant enhancements.
>
> Nevertheless, we also find that jointly performing the tracking and segmentation in our framework is useful for the segmentation task as well, achieving improved results than powerful baselines, such as SAM-PT. These segmentation baselines, on the other hand, do not provide point tracking functionality.
>
> We believe the impact of our finding is not limited to helping a tracking task, but can also inspire new methods for video segmentation by considering a synergistic effect between the segmentation and different tasks.
>
> ---
>
> **[Q1] Additional FLOPs comparison in a different baseline**
>
> Since our framework is a plug-in for a point tracking model, directly comparing FLOPs of our main TrackIME (incorporated with TAPIR) and other baseline (e.g., CoTracker) would not be a lateral comparison.
>
> Nevertheless, we provide the additional FLOPs results for CoTracker processing 64 frames in three different input dimensions: 384, 768, and 1080 pixels at the shorter side, and our TrackIME incorporated with CoTracker, experimented in DAVIS-F. In a similar trend as our experiment based on TAPIR, the benefit from the larger input dimensions is limited since the baseline tracker is only optimized for low-resolution input frames (384 × 512) to meet the memory constraints, and it is non-trivial to process high-resolution inputs without fine-tuning. When incorporated with TrackIME, on the other hand, the model can better utilize high-frequency details without fine-tuning and in a computationally efficient manner, e.g., TrackIME vs. CoTracker (1080 $\times$ 1440) gives **6217G** vs. 14138G (FLOPs), **64.5** vs. 62.2 (AJ), **79.2** vs. 76.7 ($\delta^{x}_{\text{avg}}$), **88.5** vs. 86.6 (OA), respectively.
>
> We hope this answers the question and will include these results in the updated manuscript.
>
> \begin{array}{l  c |  c  c  c  }
> \text{Method} ~ (\text{Input Dim.}) & \text{FLOPs} & \text{AJ} & \delta^x_\text{avg}  & \text{OA}  \newline
> \hline
> \text{CoTracker} ~ (384 \times 512) & 2707 \text{G} & 60.8 & 76.1 & 86.0 \newline
> \text{CoTracker} ~ (768 \times 1024) & 7670 \text{G} & 62.3 & 77.8 & 87.1 \newline
> \text{CoTracker} ~ (1080 \times 1440) & 14138 \text{G} & 62.2 & 76.7 & 86.6 \newline
> \hline
> \textbf{TrackIME} ~ \text{with CoTracker} \newline (384 \times 512) & 6217 \text{G} & \mathbf{64.5} & \mathbf {79.2} & \mathbf{88.5}  \newline
>  & \newline
> \hline
> \end{array}

---

> > ### Comment · Reviewer_7a51 · 2024-08-12
> >
> > I have carefully read the rebuttal and the reviews from other reviewer, and would thanks to the authors for their efforts on the rebuttal. My concerns about contributions and FLOPs comparison are resolved, and I will update my rating accordingly.
> >
> > Some questions still remain for instance object segmentation:
> >
> > * The authors claim that 'TrackIME removes erroneous query points for segmentation caused by tracking failures on intricate object parts.' Could the authors clarify how TrackIME accomplishes this? Additionally, why would SAM-PT fail in this context—is it solely because SAM-PT conducts tracking at a lower resolution, while TrackIME operates at higher resolutions through pruning?"

---

> > > ### Author Response · Authors · 2024-08-12
> > > **Thank you very much for the response.**
> > >
> > > Dear reviewer 7a51,
> > >
> > > We appreciate your efforts in reviewing our rebuttal and consulting our discussions with other reviewers. We are glad to hear that our rebuttal addressed your concerns well!
> > >
> > > We will address your additional question in the response below.
> > >
> > > ---
> > >
> > > **Why can TrackIME consider better query points for segmentation (i.e., better point tracking results), removing tracking failures on intricate object parts?**  (Note: The query points for video segmentation in TrackIME and SAM-PT are essentially based on the point tracking results. Therefore, we will focus on discussing how TrackIME addresses the limitations in accurately tracking the query points for segmentation).
> > >
> > > In addition to allowing higher resolutions for tracking, another core advantage of TrackIME is that it can provide a more reliable point tracking result for the segmentation query point by aggregating the multiple point trajectories on the same instance (as predicted by the segmentation model), so that the effect of a potential tracking failure of a single query point can be avoided, which SAM-PT majorly suffers from.
> > >
> > > Specifically, our enhanced point tracking is accomplished by the following three main components (which are the subject of our ablation study in Table 3) as follows:
> > > - Search space pruning (to allow for higher resolutions without significant computational overhead)
> > > - Trajectory aggregation (to aggregate multiple point trajectories on the same instance)
> > > - Progressive inference (to boost the tracking performance by reinitializing pruning windows at different scales)
> > >
> > > For example, our ablation study in Table 3 shows that all these modules play an important role in avoiding tracking failures on intricate object parts. First, as pointed out by the reviewer, the search space pruning allows TrackIME to operate at higher resolutions and it improves the tracking performance on the intricate 1-pixel scale (e.g., 23.0 $\to$ 28.2 $\text{J}_1$), as well as on the average scale (e.g., 57.5 $\to$ 62.5 $\text{AJ}$). Furthermore, we emphasize that another significant gain is achieved by employing the other two components$-$the trajectory aggregation and the progressive inference (e.g., 28.2 $\to$ 35.4 $\text{J}_1$ and 62.5 $\to$ 65.3 $\text{AJ}$; we note that the gain in the intricate $\text{J}_1$ is even larger here).
> > >
> > >
> > > In this respect, the limitation of SAM-PT is not only its lower resolution for point tracking, but also its lack of structure to incorporate multiple points on the same instance and their trajectories for better point tracking. We note that SAM-PT also considers multiple points internally, but they are merely used to predict segmentation masks under occlusion scenarios, not to improve the point tracking.
> > >
> > > ---
> > >
> > > If you have any further questions or suggestions, please do not hesitate to let us know.
> > >
> > > Thank you very much,
> > >
> > > Authors

---

### Official Review · Reviewer_BSTo · 2024-07-09

**Soundness:** 3
**Presentation:** 3
**Contribution:** 3
**Rating:** 6
**Confidence:** 4

**Summary:**

This paper presents a new framework for video point tracking from instance motion. By integrating existing segmentation and point tracking base models, the performance of point tracking is significantly improved from object-by-object optimization. Ablation experiments and extensions in zero-shot video object segmentation further validate the effectiveness of the proposed framework. The paper is generally well written and organized.

**Strengths:**

Instance segmentation and motion estimation provide an overall motion prior from the global perspective, which strengthens the spatial associations for point tracking. Based on this, coupled with an iterative post-processing step to mitigate the loss of visual information due to feature space downsampling, also contributes to high-resolution point tracking.

Ablation experiments and extensions in zero-shot video object segmentation further validate the effectiveness of the proposed solution.

**Weaknesses:**

At the trajectory aggregation step (eq 3), multiple moving points are usually selected corresponding to each object, but just averaging them according to visibility is to simple. Direct averaging does not satisfy common cases such as rotations and scale changes, and even if this is used for subsequent inference, a better initial value would be valuable. A straightforward modification would be to fit the affine motion model with these selected points.

For eq 3, 8 and 12, whether the visibilities can be regarded as confidence weights needs to be further analysed, since the supervision only have a 0-1 occlusion maps. Does this address the cases where an object is occluded for a few frames and then reappears?

**Questions:**

All five methods compared in Table 1 are inconsistent with the results reported in their original papers, most notably TAPNet's AJ which was 46.6 in the original paper but 56.5 here. The reasons for these inconsistencies need to be explained. This raises doubts as to whether the significant performance gains (from 62.8 to 69.3) of the proposed method can be compared to recent competitors (e.g. 65.9 of DOT).

Dense Optical Tracking: Connecting the Dots, CVPR 2024.

L431 mentions that changing the backend of the model improves performance, which is confusing. As I understand this process donot retrain the model. Did the authors confirm that the implementation and evaluation is correct, or are there some bugs that were addressed in this process?

**Limitations:**

yes

---

> ### Author Rebuttal · Authors · 2024-08-07
>
> Dear reviewer BSTo,
>
> Thank you for your valuable feedback and comments. We appreciate your remarks on the strengths of our paper, including the enabling high-resolution point tracking, the valid experiments for ablation, and the extended results in zero-shot video object segmentation. We will address your concerns and questions and provide additional results in the response below, which we will also include in the updated manuscript.
>
> ---
> **[W1] Employing more complex motion models for the trajectory aggregation**
>
> Thank you for the great suggestion. Although more complex motion models can satisfy cases such as rotations and scale changes, we find only marginal changes compared to our simple design, when applied to the trajectory aggregation step (Equations 3 and 4). For example, we provide the results employing an affine motion model (i.e., $T_t = A_t * T_{t-1} + b_t$) [R1] by fitting the least-square solution for $A_t$ and $b_t$ using the baseline tracking results, experimented on DAVIS-F benchmark.
>
> We would like to emphasize that the trajectory aggregation is not required to predict accurate motion, since its purpose is only to determine the center of pruning windows. Even if the center escapes out of the object due to complex motions, it still suffices as long as the pruning window can contain the query point.
>
> |     Method     |       $J_1$ |  $\text{AJ}$  | $\delta^x_1$ | $\delta^x_{\text{avg}}$ | $\text{OA}$  |
> |------------|:--------:|:------:|:------:|:------:|:------:|
> | TrackIME (w/ Affine motion) |  35.3 | 65.1 |48.0 | 78.4 | 85.7 |
> | **TrackIME (default)** |  **35.4** | **65.3** | **48.2** | **78.6** | **86.5** |
>
> [R1] Li, et al. "An efficient four-parameter affine motion model for video coding." IEEE Transactions on Circuits and Systems for Video Technology 28.8 (2017): 1934-1948.
>
> ---
>
> **[W2] The use of visibilities as the confidence weights in Equations 3, 8, and 12.**
>
> Thank you for acknowledging the importance of confidence weights. We first would like to clarify that our strategy combines both the hard 0-1 visibility predictions as well as the confidence weights (e.g., see $\mathbf{o}_t \geq 0.5$ in Equations 3 and 8). This strategy effectively mitigates potentially erroneous confidences by the false positives, since our method tends to demonstrate a high precision (the portion of true positives) for the visibility classification, e.g., we get  93.7% @ threshold 0.5 in DAVIS-F. Our strategy is valid as far as a sufficient number of visible tracking points are available. For the cases where an object is occluded for a few frames and then reappears, our framework can maintain the number of tracking points via the point re-sampling, even if the visibility classifier fails to predict the reappearance.
>
> To further support the validity of our strategy, we measure the average confidence of the true positives and the false positives and find 0.902 (true positives) and 0.737 (false positives), so the remaining false positives would be penalized through the weighted aggregation. We also provide additional ablation, where we force equal weights in the aggregation, experimented in DAVIS-F. For example, we find 1.3 points improvement in $\text{AJ}$ by using our strategy.
>
> |   Method  |    $J_1$ |  $\text{AJ}$  | $\delta^x_1$ | $\delta^x_{\text{avg}}$ | $\text{OA}$  |
> |------------|:--------:|:------:|:------:|:------:|:------:|
> | TrackIME (equal weights) |  34.9 | 64.0 | 47.9 | 78.5 | 86.3 |
> | **TrackIME (default)** |  **35.4** | **65.3** | **48.2** | **78.6** | **86.5** |
>
> ---
>
> **[Q1] Explanations for the inconsistent results with the baselines’ original reports**
> - **Inconsistent TAPNet results in Table 1**
>
> This inconsistency is mainly caused by the difference in the image backbone provided by the official checkpoint we utilized to reproduce TAPNet and TAPIR; it provides an updated ResNet18 backbone, instead of TSM-ResNet18 used by TAPNet in the original paper. Otherwise, the tracking method is the same as TAPNet (i.e., cost volumes without PIPs iterations).
>
> Nevertheless, we additionally provide the results based on the checkpoint with the original TSM-ResNet18 image backbone (which we find in the recent update of the official repository). When experimented with DAVIS-F, DAVIS-S, and Kinetics benchmarks, we find TrackIME keeps demonstrating significant gains, e.g., 32.8 $\to$ 47.0 AJs (14.2 points) in DAVIS-F. We will include these discussions and results in the updated manuscript.
>
> \begin{array}{l  c  c  c  c  c | c  c  c  c  c | c  c  c  c  c}
> &  & & &  \llap{\text{DAVIS}-\text{F}} & & & & & \llap{\text{DAVIS}-\text{S}} & & & & & \llap{\text{Kinetics}~~~~} \newline
> \text{Method} & J_1 & \text{AJ} & \delta^x_1 & \delta^x_\text{avg} & \text{OA} & J_1 & \text{AJ} & \delta^x_1 & \delta^x_\text{avg} & \text{OA} & J_1 & \text{AJ} & \delta^x_1 & \delta^x_\text{avg} & \text{OA} \newline
> \hline
> \text{TAPNet (TSM-ResNet)} & 5.8 & 32.8 & 11.1 & 48.4 & 77.6 & 6.7 & 38.4 & 12.6 & 53.4 & 81.4 & 8.1 & 38.3 & 14.4 & 52.5 & 79.3 \newline
> +~\textbf{TrackIME} & \mathbf{18.7} & \mathbf{47.0} & \mathbf{28.6} & \mathbf{60.6} & \mathbf{80.9} & \mathbf{21.5} & \mathbf{50.8} & \mathbf{32.4} & \mathbf{63.8} & \mathbf{81.4} & \mathbf{17.0} & \mathbf{44.5} & \mathbf{26.3} & \mathbf{57.9} & \mathbf{80.4} \newline
> \hline
> \end{array}
>
> - **The effect of library/hardware-dependent numerical characteristics**
>
> Since TrackIME is a plug-in to all baselines, we reproduced all results in our system for fair comparisons. This induces minor inconsistencies due to library/hardware-dependent numerical characteristics, even if the same model is used. For example, the improved performance of TAPIR (mentioned in L431) is caused by the different characteristics between JAX and PyTorch. We note that such inconsistencies are also observed in other references, e.g., $\delta^x_{\text{avg}}$ for PIPS2 on DAVIS-F: 69.1 (reported by CoTracker paper); 70.6 (reported by PIPS2 git repository); 69.4 (reported by our paper).

---

> > ### Comment · Reviewer_BSTo · 2024-08-09
> >
> > Thanks to the authors for their detailed response. I am very glad to see that the authors have provided additional experiments with a more reasonable motion model, which is essential to improve this paper. However, this raises new doubts about the experimental results; why did the theoretically better initial motion rather lead to worse final estimates? This may require a justifiable explanation before updating it to the paper.
> >
> > I have read the discussion between the authors and reviewer jQxW about the results. From my side, it is enough that the authors confirm that the reproduce settings are consistent and the comparison is fair. However, these differences lead to a potential problem in direct numerical comparisons with non-listed other methods. So clearly stating these in the paper is necessary, and the authors promise to do so. If the authors could open source the corresponding test code, it could help the community to build a comprehensive benchmark.

---

> > > ### Author Response · Authors · 2024-08-11
> > > **Thank you very much for the response.**
> > >
> > > Dear reviewer BSTo,
> > >
> > > We are happy to hear that our rebuttal addressed your questions and concerns well, and we also appreciate your efforts to additionally consult the discussion between the reviewer jQxW and the authors.
> > >
> > > We will address your additional concerns in the response below, which we will also include in the updated manuscript.
> > >
> > > ---
> > >
> > > **Why does replacing the aggregation with the affine motion model not improve the point tracking results?**
> > >
> > > We first would like to clarify that the affine motion model does not necessarily provide better initial motion (i.e., the instance trajectory for the pruning windows). For example, measuring the normalized $L^2$ distance between the estimated window centers with respect to the ground truth query trajectory yields 0.4227 (our aggregation) vs. 0.4294 (the affine motion model). In consequence, the final point tracking performance can be worse when the affine motion model is employed to predict the pruning windows.
> > >
> > > ---
> > >
> > > **Then, why does the affine motion model not provide better initial motion?**
> > >
> > > This is because the affine motion cannot be optimized with respect to the optimal trajectory of the pruning windows, but only with respect to the individual point trajectories within the instance predicted by the point tracker (e.g., TAPIR). If the affine motion model were to be fitted with the ground truth (GT) window centers (i.e., the ground truth query trajectory), the final tracking performance could be superior to that of our aggregation, as demonstrated in the table below. However, this is not a feasible objective, as the ground truth is not known a priori.
> > >
> > > Nevertheless, we believe that both our aggregation and the affine model provide comparable outputs; they are simply different methods of estimating the instance trajectory given the multiple points on the same instance. In fact, the differences in their results are very marginal when compared to our ablation study in Table 3, which replaces the aggregation with the single-point prediction by a query point. The following table presents the result for convenience.
> > >
> > > |     Method     |       $J_1$ |  $\text{AJ}$  | $\delta^x_1$ | $\delta^x_{\text{avg}}$ | $\text{OA}$  |
> > > |------------|:--------:|:------:|:------:|:------:|:------:|
> > > | TrackIME (single point) |  28.3 | 62.6 | 41.2 | 75.6 | 84.9 |
> > > | TrackIME (affine motion) |  35.3 | 65.1 | 48.0 | 78.4 | 85.7 |
> > > | TrackIME (default) |  35.4 | 65.3 | 48.2 | 78.6 | 86.5 |
> > > | TrackIME (affine motion w/ GT) |  **36.2** | **67.2** | **48.6** | **79.4** | **88.5** |
> > >
> > > ---
> > >
> > > **Additional remarks on instance trajectory estimation**
> > >
> > > We strongly agree that considering and testing different methods for the instance trajectory estimation is important for both the performance and the safety in the application, even though our method has provided reasonable results in our domain. We thank the reviewer for encouraging us to further acknowledge this point, which we are happy to include in the updated manuscript.
> > >
> > > ---
> > >
> > > **Commentary on the reproduced results**
> > >
> > > To avoid a potential problem with direct numerical comparisons with other methods in the future, we will ensure that the test code is released as well. We believe it can be a useful addition to the community, providing more options for implementing and testing point tracking models.
> > >
> > > ---
> > >
> > > If you have any further questions or suggestions, please do not hesitate to let us know.
> > >
> > > Thank you very much,
> > >
> > > Authors

---

> > > > ### Comment · Reviewer_BSTo · 2024-08-12
> > > >
> > > > Thanks to the authors' detailed replies, I have no additional questions. I will carefully consider these issues and make a final rating.

---

> > > > > ### Author Response · Authors · 2024-08-13
> > > > > **Thank you for your response!**
> > > > >
> > > > > Dear reviewer BSTo,
> > > > >
> > > > > We appreciate your efforts in reviewing our paper and actively participating in the discussion.
> > > > >
> > > > > Due to your valuable and constructive suggestions, we do believe that our paper is much improved.
> > > > >
> > > > > Thank you very much,
> > > > >
> > > > > Authors

---

### Official Review · Reviewer_jQxW · 2024-07-26

**Soundness:** 4
**Presentation:** 4
**Contribution:** 4
**Rating:** 8
**Confidence:** 5

**Summary:**

This paper tackles the problem of point tracking, where the task is to track the movement of a single point in a video. Point tracking has experienced a fairly recent deep learning revival starting with PIPs [22 in paper ref], which was inspired by a handcrafted method named Particle Video from Sand and Teller [1], with more recent follow-ups such as TAPIR [6 in paper ref], Omnimotion [8 in paper ref], and Cotracker [7 in paper ref].

Specifically, this paper tackles a key challenge faced by these contemporary point tracking models: the computational burden introduced by the cost volume operation, which is a correlation between the pointwise feature being tracked and the feature maps of all video frames in the current window being processed. This operation needs to be performed for each point being tracked, resulting in tensors as large as B x T x N x C x H x W (batch size, window length, number of points being tracked, num feature channels, video height, video width) being processed. In response, these models spatially downsample the feature space by up to a factor of 4 (in Cotracker's case) and 8 (in PIPs' case), which results in a decimation of high frequency information that can be useful in tracking points. As a consequence, these models experience a large reduction in tracking accuracy due to the downsampling.

The authors of this paper propose an alternative approach. Instead of downsampling the feature space, intelligently crop each frame to the most important region in the frame, i.e., somewhere around the point being tracked. The authors introduce a framework that takes advantage of a pretrained instance segmentation model (SAM [1 in paper ref] in this case) to segment the instance upon which the queried point is placed, to use the segmentation mask to query semantically neighbouring points, to track all these points using a pretrained tracker (TAPIR, PIPs, Cotracker, etc.), to aggregate all trajectories to a single instance trajectory, to crop each video frame about each instance trajectory point, and to re-do tracking of the original queried point in this pruned space. They even show how this can be performed recursively / progressively, where the entire process can be repeated to get an even more refined trajectory estimate. The authors also show how their framework can act as an accurate video object segmentation model, where the synergy between the point tracking and the segmentation model allows both to improve on their respective tasks (tracking and segmenting). They show how this can exceed the zero-shot video object segmentation performance (in the DAVIS dataset) of SAM-PT (a version of SAM that relies on a point tracking model to enable zero-shot video object segmentation) and various class-based video object segmentation models.

The authors evaluate their framework through exhaustive experimentation, with four main experiments:
* Evaluating point tracking performance for dynamic objects (i.e., testing on TAP-Vid-DAVIS)
* Testing the generalizability of their framework on different point tracking models.
* An ablation study where they ablate different components of their model (search space pruning using tracked query point vs. instance trajectory, and the recursive pruning scheme).
* Zero-shot video object segmentation.

Overall, the authors show substantial gains on standard benchmarks and demonstrate the synergistic effectiveness of using a pretrained instance segmentation model with a pretrained point tracker. They even show that there is no net computational efficiency loss in the additional FLOPS introduced by the segmentation model and the recursive process, since the tracking accuracy gains far outweigh the additional compute costs.

[1] Sand, P., Teller, S. Particle video: Long-range motion estimation using point trajectories. In CVPR 2006.

**Strengths:**

* Originality:
    * One might point to SAM-PT as a similar idea, but as the authors dutifully pointed out in the paper, it is a different approach. SAM-PT uses a point tracker to (effectively) track the movement of an instance, and when combined with the instance segmentation capabilities of SAM, this effectively results in a video object segmentation model in a zero-shot manner. In contrast, this paper introduces a novel approach to improving point tracking accuracy (the other way around compared to SAM-PT) where SAM is used to restrict the tracking search space, allowing them to track without downsampling the feature space spatial dimensionality and demonstrating substantial improvements by doing so. No other tracking model, as far as I'm aware, have introduced anything close to this method.
    * Related works have been adequately cited and the paper is very clear in how it differs from prior methods.
* Quality:
    * Claims are well supported by a thorough analysis on the results of exhaustive experimentation.
    * Authors are careful and honest about evaluating both the strengths and weaknesses of their framework.
* Clarity:
    * The paper is concisely written. At no point did I have trouble understanding what the authors were trying to communicate, whether it was through their math or vocabulary. I'd like to commend the authors in the care taken to introduce each concept. For example, I enjoyed their clarification of mathematical notation in L67-70 and their concise description of the benchmark metrics in L225-233.
    * The tables and their captions are well-presented, clearly showing the gains of their framework.
    * All experiment parameters necessary to recreate results have been listed, explained, and motivated both in the main manuscript and in the appendix.
    * Overall, just about every question I can think of as I was reading the paper was almost immediately answered as I kept reading. This, to me, is an indication of a well thought out paper that flows nicely.
* Significance:
    * Definitely the most impressive part of the paper: the results. They are significant in two ways, in my opinion: 1) The framework improves on the state of the art across almost all benchmarks and with all the backbone tracking models they used; 2) This framework is extremely flexible in that you can mix and match any pretrained tracker and pretrained instance segmentation model. It's not tied to a specific model on either the tracking side or instance segmentation side.

**Weaknesses:**

* Originality:
    * No weaknesses.
* Quality:
    * No weaknesses.
* Clarity:
    * No major weaknesses, but I do have some suggestions and questions relating to clarity. I have provided these in the Questions section below.
* Significance:
    * No weaknesses.

**Questions:**

* L128: I suggest sharing the threshold value here instead of saying "is set much smaller than the standard 0.5". It seems like an unnecessary obfuscation.
* L185-195: I strongly suggest mentioning the ablation study here, as it is one of your major experiments.
* Table 3: I suggest mentioning in the caption that the tracking backbone is TAPIR. I understand this is mentioned in a couple of places in the main manuscript, but it helps to remind the reader when reading the table caption, as this piece of information can be accidentally skipped when skimming through the main manuscript.
* L274: "23.2" is supposed to be 28.2, right?
* L297-298: Can you provide some brief commentary on why TrackIME may be performing better than SAM-PT on zero-shot video object segmentation? I'm curious to know your thoughts. If insightful, it may even be useful to include it in the main manuscript.
* Table 7: Are the resolutions displayed here (excluding TrackIME since there's no training involved) the input resolutions during training or during testing?
* Figure 2: The points and the images are too small! I suggest increasing the size of the points, removing the last column of images, and increasing the size of the image grid to match the width of the caption. I liked the size of the points shown in Figure 1.
* Figure 3: The points are even harder to see here than in Figure 2 despite the images being larger! This one's an easy fix: increase the size of the points (preferably to match the size of the points in Fig 1 for consistency).

Great paper overall. Good job :)

**Limitations:**

The authors have adequately addressed the limitations of their framework and its potential negative societal impacts. Limitations have been discussed in Appendix F and Section 4, with honest and descriptive commentary. Potential negative societal impacts have been sufficiently discussed in Appendix F.

---

> ### Author Rebuttal · Authors · 2024-08-07
>
> Dear reviewer jQxW,
>
> Thank you for your valuable feedback and very detailed comments. We appreciate your remarks on the strengths of our paper, including the significance of the method, exhaustive experimentation, and clear presentation. We will address your concerns and questions in the response below.
>
> ---
>
> **[Q1] Commentary on why TrackIME may be performing better than SAM-PT on zero-shot video object segmentation**
>
> The key advantage of TrackIME is removing erroneous query points for segmentation caused by the tracking failure on intricate object parts, and enabling even finer query points for segmentation, e.g., demonstrating outstanding results in 1-pixel thresholds (e.g., Table 1), which is possible due to the pruning structure in our framework to maintain the high-frequency information. Although SAM-PT contributes to enhancing the segmentation from the point tracking task, their performance is still bounded by low-resolution tracking and suffers from tracking failures.
>
> Suppose SAM could provide perfect segmentation for a given query, and the point tracking model could provide perfect tracking and occlusion predictions up to their resolutions. Nevertheless, the tracking resolution would impose an upper bound for the video object segmentation performance because the query points in the fine-grained object parts suffer from failure modes in lower resolutions.
>
> We would like to attribute the better performance demonstrated by TrackIME to this point.
>
> ---
>
> **[Q2] Table 7: Are the resolutions displayed here (excluding TrackIME since there's no training involved) the input resolutions during training or during testing?**
>
> The resolutions for TAPIR models in Table 7 are the inference resolutions, where we utilized the official checkpoint trained in the 256 x 256 resolution. We will clarify this point in the updated manuscript.
>
> ---
>
> **[Q3] L274: "23.2" is supposed to be 28.2, right?**
>
> Yes. We will correct this typo in the updated manuscript.
>
> ---
>
> **[Q4] Editorial comments**
>
> Thank you for the constructive editorial comments. In the updated manuscript, we will provide hyperparameters more clearly (e.g., $r=0.10$ for the segmentation), revise the presentation to mention our ablation studies in the introduction of Section 4 and that our tracking backbone is TAPIR in Table 3, and revamp the drawings in Figures 2 and 3.

---

> > ### Comment · Reviewer_jQxW · 2024-08-08
> > **Mostly pleased but just one more concern....**
> >
> > **[Q1] Commentary on why TrackIME may be performing better than SAM-PT on zero-shot video object segmentation**
> > >  their performance is still bounded by low-resolution tracking and suffers from tracking failures.
> >
> > Ah! Of course, that makes sense. I would suggest making a brief mention about this fact in the main manuscript, or at least in the supplemental, to ease any qualms one might have.
> >
> > ---
> >
> > I've read your responses to my questions and I'm happy to see that you'll be including my suggestions in your revised copy.
> >
> > ---
> >
> > On a side note, Reviewer BSTo raised a good point that I forgot to bring up with regards to the inconsistencies in reported results from the baselines and I wanted to give a comment about that. First, I'm glad to see that you've addressed the concerns regarding TAPNet, but you should also address the inconsistencies with the other four approaches. Personally, I'm quite familiar with the literature (and code), and so I understand that inconsistencies can arise due to small things like the filtering algorithm used when resizing DAVIS images, the version of PointOdyssey that was trained on (in the case of PIPs++), etc. For example, I know that PIPs++' reported result in the PointOdyssey paper is much lower than the reported result in its github repo, which is close to what you reported in Table 1. I also know that your CoTracker result is very close to what was reported in the CoTracker paper. Despite this, you absolutely should explain these differences and why they came about, lest you lose the reader's trust.
> >
> > On that note: why are there differences in your baseline results compared to the original reported results in their respective papers?

---

> > > ### Author Response · Authors · 2024-08-09
> > > **Thank you very much for the response**
> > >
> > > Dear reviewer jQxW,
> > >
> > > We are happy to hear that our rebuttal addressed your question well.
> > >
> > > We will also address your additional concerns and questions in the response below, which we will also include in the updated manuscript.
> > >
> > > ---
> > >
> > > **The reasons for the differences in our baseline results compared to the original reports**
> > >
> > > We are grateful for your genuine side notes, which we find very helpful in elucidating the cause of the inconsistencies in our baseline results, such as the influence of the image resize filtering and the checkpoint version on the results. As detailed in L200-202 of the manuscript, we applied the same resize filtering function, the standard bicubic filter, to every model using the default input resolution of each model. As the reviewer pointed out, our choice of resizing filter might differ from those used in the original codes and could have caused small inconsistencies in the results.
> > >
> > > Finally, we would like to gently point out that in our response to reviewer BSTo, we mention the effect of library/hardware-dependent numerical characteristics in the discrepancy in the results for all baselines. It is also worth noting that reproducing exactly the same results for OmniMotion is hindered by the stochastic nature of the optimization process, and by the fact that the default hyperparameters and the choice of activation functions in their official repository are different from those considered in their published paper (according to their Github bulletin "Issues"). Nevertheless, we believe that the effectiveness of TrackIME is intact, orthogonal to this issue.
> > >
> > > ---
> > >
> > > **Comparison to the original reports**
> > >
> > > Although the degree of the inconsistency is relatively small given the significant gain in tracking accuracy by incorporating TrackIME into the baselines, we strongly agree that clarifying these differences and explaining their causes is critical to the credibility of our work. In the updated manuscript, we will also include the reported results from the original papers, and provide detailed explanations for all factors that have caused the inconsistencies. For example, the average tracking accuracies (i.e., $\text{AJ}$ and $\delta^{x}_{\text{avg}}$) in DAVIS-F for the baselines that provide the official checkpoints (i.e., TAPNet, PIPS2, CoTracker, and TAPIR), could be presented as follows. We note that the checkpoint for TAPNet with TSM-ResNet18 (the one we mention in our response to the reviewer BSTo) is used.
> > >
> > > \begin{array}{l  c  c}
> > > \text{Method} & \text{AJ} & \delta^x_\text{avg}  \newline
> > > \hline
> > > \text{TAPNet (original paper)} & 33.0 & 48.6 \newline
> > > \hdashline
> > > \text{TAPNet (our reproduction)} & 32.8 & 48.4 \newline
> > > +\textbf{TrackIME} & \mathbf{47.0} & \mathbf{60.6} \newline
> > > \hline
> > > \text{PIPS2 (original paper)} & - & 63.5 \newline
> > > \text{PIPS2 (official repository)} & - & 70.6 \newline
> > > \hdashline
> > > \text{PIPS2 (our reproduction)} & 46.6 & 69.4 \newline
> > > +\textbf{TrackIME} & \mathbf{50.3} & \mathbf{74.0} \newline
> > > \hline
> > > \text{CoTracker (original paper)} & 62.2 & 75.7 \newline
> > > \hdashline
> > > \text{CoTracker (our reproduction)} & 60.8 & 76.1 \newline
> > > +\textbf{TrackIME} & \mathbf{64.5} & \mathbf{79.2} \newline
> > > \hline
> > > \text{TAPIR (original paper)} & 56.2 & 70.0 \newline
> > > \hdashline
> > > \text{TAPIR (our reproduction)} & 57.5 & 70.5 \newline
> > > +\textbf{TrackIME} & \mathbf{65.3} & \mathbf{78.6} \newline
> > > \hline
> > > \end{array}
> > >
> > > ---
> > >
> > > If you have any further questions or suggestions, please do not hesitate to let us know.
> > >
> > > Thank you very much,
> > >
> > > Authors

---

### Author Rebuttal · Authors · 2024-08-07

Dear reviewers and AC,

We sincerely appreciate your valuable time and effort spent reviewing our manuscript.

As the reviewers highlighted, we believe our paper provides a novel approach that incorporates point tracking models with the pre-trained instance segmentation (all reviewers). This approach provides significant performance improvements (jQxW, BsTo) in a computationally efficient manner (jQxW, 7a51), followed by a clear presentation (jQxW, BsTo).

We appreciate your constructive comments on our manuscript.

We strongly believe that TrackIME can be a useful addition to the NeurIPS community, in particular, due to the enhanced manuscript by reviewers’ comments helping us better deliver the effectiveness of our method.

Thank you very much!

Authors

---

### Decision · Program_Chairs · 2024-09-25

**Decision:**

Accept (spotlight)

**Comment:**

This paper presents a method to improve the performance and computational efficiency of point trackers, by adding awareness of the objects that the points belong to, via a pre-trained segmentation model. Despite using off-the-shelf models (without finetuning them), the method achieves substantial improvements in point tracking accuracy, across a wide array of base models, and delivers state-of-the-art zero-shot video segmentation as well. The reviews were initially leaning positive, and became more confident after the rebuttal, arriving at 1x strong accept, 1x weak accept, 1x borderline accept. The reviewers appreciated the core idea, the results, and the presentation, and the AC agrees. The authors are encouraged to incorporate the work done during the rebuttal, to strengthen the paper further. The AC recommends acceptance.